# Sedentary Time, Physical Activity, and Sleep Duration: Associations with Body Composition in Fibromyalgia. The Al-Andalus Project

**DOI:** 10.3390/jcm8081260

**Published:** 2019-08-20

**Authors:** Blanca Gavilán-Carrera, Pedro Acosta-Manzano, Alberto Soriano-Maldonado, Milkana Borges-Cosic, Virginia A. Aparicio, Manuel Delgado-Fernández, Víctor Segura-Jiménez

**Affiliations:** 1PA-HELP “Physical Activity for Health Promotion, CTS-1018” Research Group, Department of Physical Education and Sports, Faculty of Sport Sciences, University of Granada, 18011 Granada, Spain; 2Sport and Health University Research Institute, University of Granada, 18016 Granada, Spain; 3SPORT Research Group (CTS-1024), CERNEP Research Center, University of Almería, 04120 Almería, Spain; 4Department of Education, Faculty of Education Sciences, University of Almería, 04120 Almería, Spain; 5Department of Physiology, Faculty of Pharmacy and Institute of Nutrition and Food Technology, University of Granada, 18011 Granada, Spain; 6Department of Physical Education, Faculty of Education Sciences, University of Cádiz, 11519 Cádiz, Spain; 7Biomedical Research and Innovation Institute of Cádiz (INiBICA) Research Unit, Puerta del Mar University Hospital University of Cádiz, 11009 Cádiz, Spain

**Keywords:** accelerometry, GT3X+, obesity, weight, sleep duration, muscle mass, fatness

## Abstract

To explore the individual–independent relationships of sedentary time (ST) and physical activity (PA) (light and moderate-to-vigorous intensity (MVPA)), with sleep duration and body composition (waist circumference, body mass index (BMI), body fat percentage, and muscle mass index) in women with fibromyalgia, and to determine whether these associations are independent of physical fitness. This cross-sectional study involved 385 women with fibromyalgia. ST and PA were assessed by triaxial accelerometry, sleep duration was self-reported. Waist circumference was measured using an anthropometric tape, and body weight, body fat percentage, and muscle mass were estimated using a bio-impedance analyzer. In individual regression models, ST and sleep were directly associated with waist circumference, BMI, and body fat percentage (*β* between 0.10 and 0.25; all *p* < 0.05). Light PA and MVPA were inversely associated with waist circumference, BMI, and body fat percentage (*β* between −0.23 and −0.12; all *p* < 0.05). In multiple linear regression models, ST (*β* between 0.17 and 0.23), light PA (*β* between −0.16 and −0.21), and sleep duration (*β* between 0.11 and 0.14) were independently associated with waist circumference, BMI, and body fat percentage (all *p* < 0.05). MVPA was associated with waist circumference independent of light physical activity (LPA) and sleep duration (*β* = −0.11; *p* < 0.05). Except for MVPA, these associations were independent of physical fitness. These results suggest that longer ST and sleep duration, and lower PA levels (especially light intensity PA), are independently associated with greater adiposity, but not muscle mass, in women with fibromyalgia. These associations are, overall, independent of physical fitness.

## 1. Introduction

An unfavorable body composition can have harmful consequences, and the growing prevalence of this problem is a public health concern [1]. General obesity, abdominal obesity, and low percentage of muscle mass increase the risk of developing cardiometabolic disease and are associated with higher all-cause mortality [2,3]. An unfavorable body composition is also associated (often bi-directionally) with worse mental health [4], sleep [5], and physical function [6].

Fibromyalgia is a chronic condition associated with persistent and widespread pain, fatigue, non-restorative sleep, cognitive difficulties [7], and other problems. Women with fibromyalgia commonly have an unfavorable body composition (higher body mass index [BMI], a larger waist circumference, and a higher percentage of body fat) [8] and are at greater risk of developing cardiovascular disease compared to the general population [9]. In fibromyalgia, obesity has been associated with more severe symptomatology, lower quality of life, and reduced functional capacity [10,11]. It would therefore be interesting to know how modifiable factors potentially associated to obesity, such as lifestyle behaviors (sedentary time (ST), physical activity (PA), sleep duration, etc.) are actually related to parameters of body composition in fibromyalgia.

Observational data (mostly) suggest that, in adulthood, the association between sedentary behavior and adiposity–obesity is small or inexistent [12]. However, participating in more PA might be beneficial in terms of achieving a better body composition in the general population [13,14,15,16] according to the available ecological data. Inadequate sleep duration has also been connected to a range of adverse health outcomes [17], including obesity [5]. 

ST, PA, and sleep are co-dependent behaviors but the scant evidence linking them to obesity in patients with fibromyalgia [18,19] has typically led to their relationships being considered in isolation. In addition, there is a gap in the literature examining the link between ST, PA, and body composition in fibromyalgia. A better understanding of the combined association of these behaviors with different body composition variables may be valuable in defining appropriate weight management strategies for female patients with fibromyalgia. 

The present research explores the individual and independent relationships of ST, PA (light and moderate-to-vigorous intensity (MVPA)), and sleep duration, with body composition variables (waist circumference, BMI, percentage body fat, and muscle mass index (MMI)) in women with fibromyalgia and examines whether these associations are independent of physical fitness. 

## 2. Methods

### 2.1. Settings and Eligibility Criteria

This study was performed within the framework of the Al-Andalus project. The sampling procedures performed to obtain a representative sample size of southern Spanish women with fibromyalgia (*n* = 300) have been described elsewhere [20]. Data from the participants were collected between 1 November 2011 and 31 December 2013. Subjects were contacted via patient associations in different provinces of Andalusia, via internet advertisements and e-mail. After providing detailed information about the study procedures, written informed consent to be included was obtained from all subjects (*n* = 646). The inclusion criteria were: (i) To have been diagnosed with fibromyalgia by a rheumatologist, (ii) to meet the official 1990 American College of Rheumatology (ACR) criteria for fibromyalgia [21], and (iii) to be a woman. Participants were excluded if they had any acute or terminal illness, severe cognitive impairment, or were >65 years old (to avoid the influence of other prevalent conditions related to age deterioration). The study was reviewed and approved by the Ethics Committee of the Hospital Virgen de las Nieves (Granada, Spain).

### 2.2. General Procedure

Baseline data were collected on two different days. On the first day, the meeting of inclusion criteria was checked, and participants provided their sociodemographic and clinical data. Anthropometric data were recorded, and body composition and tender points (according to 1990 ACR criteria [21]) were assessed. Two days later, the subjects completed the Pittsburgh Sleep Quality Index questionnaire, their physical fitness was assessed, and they were provided with a hip-worn accelerometer to capture PA data over the following nine days. 

### 2.3. Outcome Measures

#### 2.3.1. Anthropometry and Body Composition

A single measurement of waist circumference (cm) was carried out using a Harpenden anthropometric tape (Holtain Ltd., Crymych, UK) placed at the midpoint between the ribs and the iliac crest. A portable InBody R20 eight-polar tactile electrode impedanciometer (Biospace, Seoul, Korea) was used to measure body weight (kg), and to estimate percentage of body fat and muscle mass. This analyzer has demonstrated accurate estimates of total and appendicular body composition [22]. A stadiometer (Seca 22, Hamburg, Germany) was used to measure height. BMI was calculated as weight (kg)/height (m) squared. MMI was calculated as muscle mass (kg)/height (m) squared.

#### 2.3.2. Sedentary Time and Physical Activity

Patients wore a GT3X+ triaxial accelerometer (Actigraph, Pensacola, FL, USA) around the hip for nine consecutive days (24 h per day except during water-based activities). Data from the first day (to avoid reactivity) and the last day (device return) were excluded from the analysis. Data were collected at a rate of 30 Hz; the epoch length was 60 s [23]. Values for seven continuous days, with a minimum of 10 h of the waking wear time per day, were required for a subject’s data to be included in the analyses. Data download, reduction, cleaning, and analysis were conducted using the manufacturer’s software (Actilife^TM^ v.6.13.3 desktop). 

Accelerometer wear time was calculated by subtracting sleeping time (obtained from a sleep diary) and the non-wear period from the total time. Bouts of 90 continuous min (30 min minimum up/down stream time for consecutive zero counts, and a 2 min skip tolerance) with no data (counts) were considered non-wear time according to the Choi algorithm [24]. Sedentary time was estimated as the time accumulated below 200 counts per minute (cpm) during wear time [25]. This cut-off point for sedentary time is irrespective of postural information (i.e., it does not allow to distinguish between different postures such as sitting/lying and standing). PA intensity (light, moderate, and vigorous) was calculated based upon recommended PA vector magnitude cut-offs: 200–2689, 2690–6166, and ≥6167 cpm, respectively [23]. The subjects had extremely low vigorous PA values (0.4 min/day); vigorous PA was therefore excluded as a behavior and moderate-to-vigorous PA (MVPA) was included instead. MVPA accumulated in bouts (bouted MVPA) was also calculated. A bout was defined as a period of ≥10 consecutive minutes of MVPA (up to 2 min below the cut-off point). 

#### 2.3.3. Sleep Duration

Sleep duration was self-reported using the item from the Spanish version of the Pittsburgh Quality Sleep Index. This questionnaire is considered a reliable instrument with a good convergent validity for measuring sleep quality (and sleep duration) among fibromyalgia patients [26]. More specifically, the patients were asked “During the past month, how many hours of actual sleep did you get at night?”

#### 2.3.4. Tenderness

Following the 1990 ACR criteria for classification of fibromyalgia [21], 18 tender points were assessed using an FPK 20 standard pressure algometer (Wagner Instruments, Greenwich, CT, USA). 

#### 2.3.5. Physical Fitness

Physical fitness was assessed using the standardized Senior Fitness Test battery [27]. These tests have proved to be reliable in women with fibromyalgia [28] and include different components of physical fitness: Flexibility, muscle strength, speed-agility, and cardiorespiratory fitness. Upper-body flexibility was assessed with the back scratch test, where the cm between (positive distance) or overlap (negative distance) of the middle fingers behind the back was recorded twice for each arm. The best scores of dominant and non-dominant side were used and the total mean was calculated. Lower-body flexibility was assessed with the chair sit and reach test. The patient, while seated with one leg extended, slowly bends forward sliding the hands down the extended leg in an attempt to touch (or pass) the toes. The number of centimeters short of reaching the toe (minus score) or reaching beyond it (plus score) was recorded. A total of two trials with each leg were measured and the best value of each leg was registered. The average of both legs was used in the analyses. Upper-body muscle strength was assessed with the arm curl test. In this test, the number of times a hand weight (2.3 kg) can be curled through a full range of motion in 30 s was registered. Patients performed the test with both hands and the average was used in the analyses. Lower-body muscle strength was assessed with the 30-sec chair stand test in which we registered the times an individual is able to rise from a sitting position to a full stand position with arms folded across chest. Cardiorespiratory fitness was assessed with the 6-min walk test that measures the distance in meters that a participant is able to walk along a 45.7 m course within 6 min. 

#### 2.3.6. Dietary Habits 

Dietary habits were self-reported through completion of a short form of a validated food frequency questionnaire [29] in which participants indicated the frequency of consumption (number of times per day, week, month, or year) of 34 foods divided by food groups: Fruit, vegetables, dairy products, fish, cereals, pulses, eggs, meat, fats, sweets, beverages, and nuts. More detailed information about the procedures and dietary habits of these participants are available in a previous study from our research group [30].

### 2.4. Statistical Analyses

The sociodemographic and clinical characteristics of the sample were summarized by descriptive statistics. Bivariate correlation and linear regression analyses were used to identify potential confounders related to sociodemographic, clinical, and dietary habits variables, among others. As a result, accelerometer wear time, age, marital status, professional status, current regular menstruation, and current consumption of alcohol were introduced as covariates in all regression models. Those involving MMI as an outcome variable were additionally adjusted for fat mass (kg) since muscle mass might also be increased in obese individuals [31]. 

The relationships between ST, PA, and sleep duration (predictors) with the measured body composition variables (outcome variables) were analyzed by univariate linear regression in separate regression models (individual associations). These models informed about the association of each behavior with the outcome of interest when considered in isolation and omitting the contribution of other behaviors (i.e., no other behaviors included in the model).

Subsequently, to test the independent associations of ST, PA, and sleep duration with body composition, multiple linear regression models were constructed including all behaviors simultaneously. These models informed about the association of each behavior with the outcome of interest when considering the contribution of other behaviors simultaneously (i.e., all behaviors included in the same model). Due to the multicollinearity problems identified between ST and light PA, two regression models were constructed for each body composition variable: (i) ST + MVPA + sleep duration + covariates and (ii) light PA + MVPA + sleep duration + covariates. For the association between sleep and body composition, only data derived from the linear regression analyses is shown; the distribution of the data was linear and the stratification of subjects into sleep duration groups (typically employed because of the U-shaped trend) returned the same results.

To examine the effect of physical fitness on the association of ST, PA, and sleep duration with body composition, a physical fitness index was included as an additional predictor in the aforementioned models. This index was created with the average of the z scores ((value-mean)/standard deviation) for each physical fitness component. 

To further analyze the influence of accumulating MVPA in bouts of at least 10 consecutive minutes, all regression models were replicated with bouted MVPA instead of MVPA.

All assumptions associated with the generalization of the results (linearity, additivity, normal distribution of residuals, homoscedasticity, independence of the errors, and multicolinearity) were reasonably met in the different regression models. 

All calculations were performed using the Statistical Package for the Social Sciences v.20.0 (IBM Corp, Armonk, NY, USA). Statistical significance was set at *p* ≤ 0.05.

## 3. Results

The final study sample (selected from among 646 potential subjects) was composed of 385 Caucasian women with fibromyalgia (age 51 ± 8 years, BMI 28.5 ± 5.4 kg/m^2^) (Appendix A). Table 1 shows their sociodemographic data and clinical characteristics.

Table 2 shows the individual associations of ST, PA intensity, and sleep duration with body composition. ST was directly associated with waist circumference (*β* = 0.247), BMI (*β* = 0.226), and percentage body fat (*β* = 0.184) (all *p* < 0.05). Light PA was inversely associated with waist circumference (*β* = −0.234), BMI (*β* = −0.215), and percentage of body fat (*β* = −0.175) (all *p* < 0.05). Sleep duration was directly associated with waist circumference (*β* = 0.117), BMI (*β* = 0.097), and percentage of body fat (*β* = 0.099) (all *p* < 0.05). MVPA was inversely associated with waist circumference (*β* = −0.167), BMI (*β* = −0.150), and percentage of body fat (*β* = −0.121) (all *p* < 0.05). Bouted MVPA was negatively associated with waist circumference (*β* = −0.120), BMI (*β* = −0.129), and with percentage of body fat (*β* = −0.103) (all *p* < 0.05).

Table 3 shows the independent associations of ST, PA intensity, sleep duration, and physical fitness with body composition. When ST, MVPA, and sleep duration were simultaneously entered into the models, ST and sleep duration were found to be independently and positively associated with waist circumference, BMI, and percentage of body fat (*β* from 0.113 to 0.230) (all *p* < 0.05). When adding physical fitness to the models, the results did not significantly change. When light PA, MVPA, and sleep duration were simultaneously introduced into the models, light PA (negatively: *β* from −0.205 to −1.55, *p* < 0.01) and sleep duration (positively: *β* from 0.113 to 0.144, *p* < 0.05) were independently associated with waist circumference, BMI, and fat percentage. MVPA was also inversely associated with waist circumference (*β* = −0.11, *p* < 0.05). When adding physical fitness to the models, the results remained similar, except for the association between MVPA and waist circumference, which became non-significant (*p* > 0.05). When the analyses were replicated with bouted MVPA, the results remained similar (Appendix A).

## 4. Discussion

The present results show that, in women with fibromyalgia, increased ST and sleep duration, and less light PA and MVPA, are individually associated with increased obesity indicators (waist circumference, BMI, and fat percentage), but not with MMI. Moreover, ST and sleep duration (directly), and light PA (inversely) were independently associated with waist circumference, BMI, and fat percentage, but not with MMI. ST showed the strongest individual and independent association with all the measured obesity indicators. In addition, MVPA and bouted MVPA were inversely associated with waist circumference independent of light PA and sleep duration, but not of ST. These results were consistent after adjusting for physical fitness—except for the relationship of MVPA with waist circumference, which became non-significant. 

Regarding the ST associations, the present results cannot be compared to those of other studies in patients with fibromyalgia: The literature contains no such reports. One study conducted in patients with rheumatoid arthritis reported ST to be strongly and positively associated with BMI [32]. In the general population, the literature is generally supportive of a small or inexistent association between sedentary behavior (assessed through different self-reported and device-based measures) and adiposity–obesity in adulthood [12], a point made in a recent systematic review of prospective studies [33]. Several studies that objectively assessed the effect of ST [15,34,35,36,37] on BMI, waist circumference, and fat mass [34,35,36] found direct weak–medium associations to exist; the present results are in agreement with these findings. Interestingly, this association was not confirmed in two other studies that made use of accelerometers, [15,37] perhaps because the methodology followed was different (different measurements of adiposity and body weight, and different data handling). The association observed in the present study between ST and total and central adiposity might be explained by three physiological mechanisms: (i) An immobilization-induced stress response due to a sedentary lifestyle [38] might promote muscle insulin resistance and atrophy, favoring the saved calories to be reallocated into fat production and storage in adipocytes; (ii) the lipid overflow-ectopic fat theory [2,3]—an excess of fat potentially involves the dysfunctional activity of the normal storage mechanism, induced by a sedentary lifestyle and excessive calorie consumption; and (iii) sedentary behavior, which implies reduced muscular activity, might contribute to poor lipid and glucose metabolism through the downregulation of the mitochondrial and lipoprotein lipase activity of the skeletal muscles [39]. Self-reported sedentary behavior has been inversely associated with muscle mass [40], but no such association was found in the present research. Since muscle mass in obese individuals may be increased [31], MMI was adjusted for fat mass in the present analyses. ST was positively associated with muscle mass before fat mass was included in the analyses; this may suggest a mediating role for fat mass in the relationship between ST and MMI. 

Light PA showed stronger individual associations with body composition variables than either MVPA or bouted MVPA (strength of the association: Weak–medium). Light PA was the only PA level associated with all obesity indicators, independent of all other behaviors. In patients with rheumatoid arthritis, several studies have suggested that total PA is inversely associated with BMI [41,42] and body fat percentage [42]. In contrast, Elkan et al. [43] found no association between total PA either with these variables or with fat free mass in such patients. A previous study in patients with rheumatoid arthritis showing a wide range of PA intensities [32] also observed a medium and inverse association of moderate—but especially of light PA (strong relationship)—with BMI. When using objective measures of PA (for the general population), the literature is mostly in agreement with the results of the present study; [34,35,36] only a few studies report a medium positive (when adjusted for MVPA) [34] or non-significant [36] association between light PA and certain body composition variables. The inverse association of light and MVPA with obesity indicators might be similarly explained by the above-mentioned pathways. Given that a greater duration and intensity of PA are determinants of greater contractile activity-induced energy expenditure [44], it might be speculated that the better body composition profile of those subjects who undertook more light PA (instead of MVPA) might be partially explained by the longer time that women with fibromyalgia commonly spend in activities of light intensity compared to more intense activity [45]. 

A positive weak–medium association was found between self-reported sleep duration and all obesity indicators, which contrasts with the findings of two early studies on patients with fibromyalgia [18,19]. The latter studies reported sleep duration, assessed via accelerometry, to be moderately and inversely associated with BMI. Most of the literature concerning the general population is supportive of a U-shaped association between sleep duration and obesity; [5] few studies have failed to find such an association [46] or found a weak–medium positive association similar to that seen in the present research [47]. However, it is difficult to make comparisons with most of these previous articles since: (i) There are notable discrepancies between self-reported sleep and accelerometer-measured values in fibromyalgia, [48] and (ii) the present participants were mainly sleep deprived (346 min/day of average sleep) whereas the duration of sleep reported in studies on the general population had a wide range. Moreover, the positive and inverse trend observed in our study might be also partially explained by dysfunctional immunometabolic responses (related to disturbances of sleep continuity [49]), and altered hormonal [50,51] and circadian responses [52,53], among others. Hence, it is necessary to further explore these associations in studies of women with fibromyalgia with higher sleep heterogeneity (i.e., with greater sleep duration), and to examine if their immunometabolic, hormonal, and circadian responses to short sleep duration are different from the general population. It has also been reported that adult women who sleep longer (exceeding the normal range) have a significantly lower skeletal muscle mass [54,55]. However, no such association between sleep duration and MMI was seen in the present research. Fat mass would appear to be a possible determinant in this lack of association. 

Interestingly, the connections between body composition and ST, light PA and sleep duration, were independent of physical fitness (strength of the associations: Weak–medium). Hence, fostering greater level of physical fitness (through progressive PA or physical exercise) might be also a complementary and/or alternative strategy to favor a better body composition profile. The idea that both PA and physical fitness may help combat obesity [56] might be extended to patients with fibromyalgia. However, the association between MVPA and waist circumference was not significant when physical fitness was included in the present models. Theoretically, physical fitness is partially determined by MVPA, and thus should explain a similar amount of variance in the statistical model. While current recommendations for PA are focused on the accumulation of a certain amount of MVPA for health benefits to be had [16], the present results agree with other evidence suggesting the beneficial role of light PA in patients with fibromyalgia [57,58]. Indeed, the present results relating body composition and bouted MVPA were echoed with respect to MVPA. Hence, promoting strategies aimed at encouraging small changes in activity might be beneficial in this population. Indeed, this “small-changes” approach might increase responsiveness in overweight individuals [59]—as is commonly the case in patients with fibromyalgia [8]. 

The present findings need to be interpreted in the light of the study’s limitations. Firstly, the cross-sectional design does not allow causal relationships to be established. In addition, only interested women took part in this study. Moreover, the GT3X+ accelerometer cannot recognize between different postures that constitute sedentary behavior (this is, sitting/lying/reclining) and ST could be overestimated since standing time may be also classified as sedentary behavior [25]. Moreover, sleep duration was self-reported. Although the potential influence of some confounders associated with diet (for instance, alcohol consumption) was analyzed and controlled, energy intake was not included. This represents the greatest limitation of the present study as obesity is determined by long-term excess energy intake above energy expenditure. The assessment of energy intake would have allowed us to better address the association between lifestyle behaviors and body composition parameters, which warrants its inclusion in future studies. In addition, we attempted to consider the combined effect of different lifestyle behaviors, yet more sophisticated approaches such as compositional data analyses [35], could accurately infer the influence of different distributions of time spent in physical co-dependent behaviors. The study is, however, benefitted by a number of strengths: (i) The large number of participants makes the sample more representative of southern Spanish women with fibromyalgia; and (ii) the use of accelerometers, a bioimpedance analyzer, and a physical fitness test battery to objectively measure ST, PA levels, body composition, and physical fitness. 

## 5. Conclusions

In conclusion, less ST and sleep, and more light PA and MVPA, are individually associated with reduced obesity indices (waist circumference, BMI, and fat percentage), but not with muscle mass. ST, sleep duration (directly), and light PA (inversely) were independently associated with obesity indicators (waist circumference, BMI, and fat percentage). Higher MVPA was associated with lower waist circumference, independent of the amount of light PA undertaken and sleep duration, but not independent of ST. These results were independent of physical fitness, except for the association of MVPA with waist circumference. While these results are consistent with the benefits of a more active lifestyle in women with fibromyalgia, the ambiguous role of sleep duration on body composition in such patients deserves further research. Future longitudinal and intervention studies objectively measuring different sedentary behaviors (e.g., sitting, lying, reclining), and considering energy intake, are warranted to confirm the present findings.

## Figures and Tables

**Table 1 jcm-08-01260-t001:** Sociodemographic and clinical characteristics of the study participants (*n* = 385).

	Mean	(SD)
**Age (years)**	51	(8)
**Marital status, *n* (%)**		
Married	295	(76.6)
With partner/without partner/divorced/widow	90	(23.4)
**Professional status, *n* (%)**		
Work full/part time	106	(27.5)
Unemployed/Retired/Housekeeper	279	(72.5)
**Educational level, *n* (%)**		
Non university degree	332	(86.2)
University degree	53	(13.8)
**Tenderness**		
Total tender points (11–18)	16.7	(2.0)
Algometer score (18–144)	43.4	(13.2)
**Current regular menstruation, *n* (%)**	123	(31.9)
**Current consumption of alcohol, *n* (%)**	137	(35.6)
**Medication consumption (yes, %)**		
Antihypertensive medication	88	(22.9)
Glycaemic lowering medication or insulin treatment	15	(3.9)
Lipid lowering medication	76	(19.7)
Analgesics	346	(89.9)
Medication for relaxation or sleep	278	(72.2)
**Sedentary time and physical activity**		
Sedentary time (min/day)	459.3	(102.9)
Light PA (min/day)	419.4	(91.7)
Moderate-to-vigorous PA (min/day)	44.5	(29.6)
Bouted moderate-to-vigorous PA (min/week)	85.1	(109.7)
Accelerometer wear time (min/day)	923.2	(76.9)
**Sleep duration (min/day)**	345.7	(88.2)
**Body composition**		
Waist circumference (cm)	90.0	(12.8)
Body mass index (kg/m^2^)	28.5	(5.4)
Body fat percentage (%)	40.0	(7.6)
Muscle mass (kg)	22.8	(3.3)
Muscle mass index (kg/m^2^)	9.1	(1.0)

PA, physical activity. Continuous variables are presented as mean (standard deviation) and categorical variables as number (percentage).

**Table 2 jcm-08-01260-t002:** Individual associations of sedentary time, physical activity intensity, and sleep duration with body composition variables.

	**Waist Circumference**					**Body Mass Index**				
	***B***	**95% CI**		***β***	**Adjusted R^2^**	***B***	**95% CI**		***β***	**Adjusted R^2^**
Sedentary time	**0.031 *****	0.018	0.043	0.247	0.145	**0.012 *****	0.006	0.017	0.226	0.102
LPA	**−0.033 *****	−0.047	−0.018	−0.234	0.136	**−0.013 *****	−0.019	−0.006	−0.215	0.095
MVPA	**−0.072 *****	−0.114	−0.031	−0.167	0.12	**−0.027 ****	−0.045	−0.009	−0.150	0.080
Sleep duration	**0.017 ***	0.003	0.031	0.117	0.108	**0.006 ***	0.000	0.012	0.097	0.069
Bouted MVPA ^b^	**−0.014 ***	−0.025	−0.003	−0.120	0.107	**−0.006 ****	−0.011	−0.002	−0.129	0.075
	**Body Fat Percentage**					**Muscle Mass Index ^a^**				
	***B***	**95% CI**		***β***	**Adjusted R^2^**	***B***	**95% CI**		***β***	**Adjusted R^2^**
Sedentary time	**0.014 *****	0.006	0.021	0.184	0.116	0.000	−0.001	0.000	−0.045	0.364
LPA	**−0.015 *****	−0.023	−0.006	−0.175	0.111	0.001	0.000	0.002	0.050	0.364
MVPA	**−0.031 ***	−0.056	−0.006	−0.121	0.101	0.000	−0.003	0.003	0.010	0.362
Sleep duration	**0.009 ***	0.000	0.017	0.099	0.099	0.001	0.000	0.001	0.045	0.363
Bouted MVPA ^b^	**−0.007 ***	−0.014	0.000	−0.103	0.114	0.000	−0.001	0.001	0.002	0.362

*B*, non-standardized coefficient; *β*, standardized coefficient; CI, confidence interval; LPA: Light physical activity; MVPA: Moderate-to-vigorous physical activity. Models were adjusted for accelerometer wear time, age, marital status, professional status, current regular menstruation, and current consumption of alcohol. ^a^ Models using muscle mass index were additionally adjusted for fat mass (kg). ^b^ MVPA (min/week) accumulated in bouts of at least 10 min. Significant associations are highlighted in bold with asterisks * *p* ≤ 0.05, ** *p* ≤ 0.01, *** *p* ≤ 0.001.

**Table 3 jcm-08-01260-t003:** Independent association of sedentary time, physical activity intensity, sleep duration, and physical fitness with body composition.

	Waist Circumference					Body Mass Index					Body Fat Percentage					Muscle Mass Index ^a^				
	*B*	95% CI		*β*	Adjusted R^2^	*B*	95% CI		*β*	Adjusted R^2^	*B*	95% CI		*β*	Adjusted R^2^	*B*	95% CI		*β*	Adjusted R^2^
Sedentary time	**0.029 *****	0.013	0.044	0.230	0.162	**0.011 *****	0.004	0.018	0.212	0.114	**0.013 ****	0.004	0.022	0.174	0.124	−0.001	−0.002	0.001	−0.051	0.364
MVPA	−0.019	−0.069	0.031	−0.043		−0.006	−0.028	0.015	−0.035		−0.007	−0.038	0.023	−0.028		−0.001	−0.004	0.003	−0.019	
Sleep duration	**0.021 ****	0.007	0.035	0.144		**0.008 ***	0.002	0.014	0.126		**0.010 ***	0.001	0.018	0.113		0.001	0.000	0.002	0.058	
Sedentary time	**0.027 *****	0.012	0.043	0.222	0.170	**0.011 ***	0.004	0.017	0.202	0.125	**0.012 ****	0.003	0.022	0.165	0.133	−0.001	−0.002	0.001	−0.051	0.362
MVPA	−0.015	−0.065	0.035	−0.034		−0.004	−0.026	0.017	−0.024		−0.004	−0.035	0.026	−0.017		−0.001	−0.004	0.003	−0.019	
Sleep duration	**0.022 ****	0.009	0.036	0.154		**0.008 ****	0.002	0.014	0.138		**0.011 ***	0.002	0.019	0.125		0.001	0.000	0.002	0.058	
Physical fitness score	**−4.074 ***	−7.835	−0.313	−0.108		**−1.991 ***	−3.622	−0.360	−0.125		**−2.645 ***	−4.935	−0.355	−0.118		−0.013	−0.284	0.258	−0.004	
LPA	**−0.029 *****	−0.044	−0.013	−0.205	0.162	**−0.011 *****	−0.018	−0.004	−0.189	0.114	**−0.013 ****	−0.022	−0.004	−0.155	0.124	0.001	−0.001	0.002	0.045	0.364
MVPA	**−0.047 ***	−0.090	−0.004	−0.110		−0.018	−0.036	0.001	−0.097		−0.02	−0.046	0.006	−0.078		0.000	−0.003	0.003	−0.004	
Sleep duration	**0.021 ****	0.007	0.035	0.144		**0.008 ***	0.002	0.014	0.126		**0.01 ***	0.001	0.018	0.113		0.001	0.000	0.002	0.058	
LPA	**−0.027 *****	−0.043	−0.012	−0.197	0.170	**−0.011 ****	−3.622	−0.360	−0.180	0.125	**−0.012 ****	−0.022	−0.003	−0.147	0.133	0.001	−0.001	0.002	0.046	0.362
MVPA	−0.042	−0.085	0.001	−0.098		−0.015	−0.017	−0.004	−0.083		−0.017	−0.043	0.010	−0.065		0.000	−0.003	0.003	−0.004	
Sleep duration	**0.022 ****	0.009	0.036	0.154		**0.008 ***	−0.034	0.004	0.138		**0.011 ***	0.002	0.019	0.125		0.001	0.000	0.002	0.058	
Physical fitness score	**−4.074 ***	−7.835	−0.313	−0.108		**−1.991 ***	0.002	0.014	−0.125		**−2.645 ***	−4.935	−0.355	−0.118		−0.013	−0.284	0.258	−0.004	

*B*, non-standardized coefficient; *β*, standardized coefficient; CI, confidence interval; LPA: Light physical activity; MVPA: Moderate-to-vigorous physical activity. Models were adjusted for accelerometer wear time, age, marital status, professional status, current regular menstruation, and current consumption of alcohol. Significant associations are highlighted in bold with asterisks * *p* ≤ 0.05, ** *p* ≤ 0.01, *** *p* ≤ 0.001. ^a^ Models using muscle mass index were additionally adjusted for fat mass (kg).

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
