# Peer review of "Sedentary Time, Physical Activity, and Sleep Duration: Associations with Body Composition in Fibromyalgia. The Al-Andalus Project"

_jcm, 2019, doi:10.3390/jcm8081260_

Round 1

Reviewer 1 Report

Please see attached document for comments.

Reviewer 2 Report

This is an interesting paper, well written and well structured. The authors explored associations between physical activity, sedentary time and sleep duration and body composition in people with fibromyalgia.

I would like to provide the following comments:

In the Discussion section (lines 207 – 208), the authors wrote:

‘Unfortunately, the present results cannot be compared to those of other studies in patients with fibromyalgia: the literature contains no such reports.’ If this is the case, thier study is filling a research gap, which should be flagged in the Introduction section.

On line 66, they mentioned that ‘ST, PA and sleep are co-dependent behaviours’, but this was not discussed in the Discussion section. It would be valuable to have a discussion about the potential of using compositional data analysis (CoDA) approaches to deal with this co-dependency in the future. In Section 2.1, please clarify when the study data were collected. Could you please add a sentence to explain why patients were excluded if they were >65 years old? Could you add a sentence to explain why 9 days of data were collected but only 7 days of data were used, please? Could you please explain in the paper what individual and independent associations are, so that a lay person could understand? The sentence on lines 206-207 is a repeat from the paragraph above it. There is a typo on line 64: ‘Inadequate sleep duration has been also been connected …’.

Round 2

Reviewer 1 Report

I praise the authors for their thorough and open approach to the initial review. I have some relatively minor comments that remain.

Line 48- inadequate is still not appropriate, unfavourable as used in subsequent lines is acceptable.

Line 122- as well as defining the accelerometer cut point for sedentary time you should clarify that this is irrespective of postural information.

Line 201 and table – Age is still expressed to 1 decimal place, this should be expressed as a whole number.

Line 335 – The lack of energy intake information is the biggest weakness of this work – yet this limitation is buried in the section on limitations. It should be given greater prominence within this section and the authors should consider including measurement of this aspect as a future direction in the conclusion.
